# A Two-Way Split-Step Wavelet Scheme for Tropospheric Long-Range Propagation in Various Environments

**Thomas Bonnafont** [1,†] , **Othmane Benhmammouch** [2,†] **and Ali Khenchaf** [1,*,†]

1   Lab-STICC, UMR, CNRS 6285, ENSTA Bretagne, 29806 Brest, France; thomas.bonnafont@ensta-bretagne.fr
2   Department of Applied Mathematics, Computer Science and Smart Systems, International University of Casablanca, Bouskoura 50169, Morocco; othmane.benhmammouch@gmail.com
*   Correspondence: ali.khenchaf@ensta-bretagne.fr
†   These authors contributed equally to this work.

**Abstract:** In the context of improving the dimensioning of observation and telecommunication, the characterization of the propagation canal is very important. Thus, accurate models of propagation phenomenona in their environment and above a rough surface (maritime or terrestrial) are of major interest for many applications (such as radar, communications, and teledetection). To provide solutions to this problem, in this paper, we propose a fast, memory-efficient, and accurate asymptotic method for 2D tropospheric propagation for a large band of frequency that accounts for relief, as well as ground composition and roughness. This latter is a two-way split-step wavelet scheme with an intrinsic stopping criterion. For overseas propagation, roughness effects are considered through a hybrid method. A complete theoretical comparison with SSF in terms of memory and time efficiency is proposed. Simulations in various environments (ground, sea, and snow), as well as different frequencies (UHF, S, and X-band) are performed to validate the method and highlight its advantages. To highlight the interest of the developed methodology, this latter is applied to different real-life applications, such as the prediction of radar coverage and the optimization of an antenna location.

**Keywords:** tropospheric propagation; split-step method; wavelet; rough surface; atmospheric duct

## 1. Introduction

Accurate modeling of tropospheric long-range propagation is important for many applications in surveillance, communication, and remote sensing, for instance, the optimization of an antenna position based on the location conditions. This is also particularly important for predicting the coverage of new systems or the impact of man-made structures on the coverage of existing systems (e.g., the impact of solar panels or wind turbines on system performance) [1]. Fast and accurate modeling of the electromagnetic wave propagation is also important for inverse problems such as refractivity from clutter (RFC) [2,3] or radio-occultation [4,5]. In this context, one must consider different interactions of electromagnetic waves with the propagation medium, such as relief, atmospheric ducts, or rough surfaces.

Due to the mesh-size limitation, rigorous methods are not suitable. Indeed, the discretization steps must be of the order $\lambda/10$ for methods such as the finite difference time domain [6], the method of moments [7], or the finite element [8]. Assuch, we use an asymptotic method. Ray-based methods [9] could be thought of, since they are accurate and model a wide range of physical phenomenon, such as the diffraction effects [10], but in our case they are limited due to the caustic problem, the shadow area problem, or the number of rays needed to account for all the physical phenomenon [11,12]; whereas the relief is a limitation for the Gaussian beam method [13,14]. Thus, we use an asymptotic model based on the parabolic wave equation (PWE) [15–17], which is adapted here, and commonly used in this context. As a matter of fact, the effects of the refraction, terrain, relief, and diffraction are considered in this model [16,17]. This latter is based on a simplification of the

Helmholtz equation by only considering the forward propagation in a paraxial cone [16,17]. Therefore, no backward propagation is introduced in the model.

The two main computational schemes commonly used to solve the PWE are either a finite difference (FD) [18] one or the split-step Fourier (SSF) method [16,17]. The latter is widely used in our context, since it allows wide steps in the propagation direction. Indeed, with the FD scheme, a mesh size of $\lambda$ [17] is required, while, with SSF, the step size in the propagation direction is of order the $100\lambda$ [17]. In this scheme, the propagation is performed in two steps. First, the field is propagated through a layer of free space in the spectral domain. Second, the effects of refraction are considered in the spatial domain through a phase screen [17]. The discrete mixed Fourier transform [19,20] allows us to take into account impedance ground conditions. The relief can also be considered with different methods [17,21,22], such as the staircase model [17]. To avoid spurious solutions, a self-consistent algorithm has been proposed [23]. Furthermore, to overcome the problem of the backward propagation, a two-way SSF [24–26] algorithm has been introduced, allowing us to precisely consider multiple reflections and multi-path effects.

Recently, a wavelet-based scheme has been developed in 2D [27–30] and 3D [31,32] to improve the memory efficiency of the method and to accelerate it to propagation. This latter follows the same steps as SSF, but the free-space propagation step is performed in the wavelet domain instead of the Fourier one. Indeed, the lower complexity of the fast wavelet transform (FWT) [33] over the fast Fourier transform (FFT) and the compression performed on the wavelet coefficients allow us to obtain an efficient method [29].

The objective of this article is to propose a fast, memory efficient, and reliable asymptotic model for 2D tropospheric electromagnetic wave propagation for a large band of frequencies that accounts for relief, ground composition and roughness, and refraction. The contributions are thus threefold. First, a two-way SSW scheme is proposed, which departs from the two-way SSF method [25,26], indeed the stopping criteria are shown to be intrinsic here. Second, the hybrid approach proposed in [34] to take into account the ground roughness effects is introduced into SSW. Third, numerical tests are proposed to validate and test the method related to practical problems, such as a radar coverage or the optimization of an antenna location.

The remainder of this article is organized as follows. Section 2 introduces the method. Firstly, the model and the discretization are explained. Secondly, a brief reminder of the 1D discrete wavelet transform is performed. Thirdly, an overview of the SSW scheme is provided. First, SSW is described for solving the one-way PWE. Second, the method is generalized for the two-way case. Third, the hybrid approach to consider the rough sea surface is introduced. Finally, a comparison between SSF and SSW in terms of the complexity and of the memory usage is proposed. Section 3 is devoted to the numerical experiments in various conditions. Section 4 concludes the paper and discusses the advantages and limitations of the proposed method. Finally, perspectives for future works are outlined.

## 2. Materials and Methods

### 2.1. Description of the Propagation Model and Discretization

In this section, we first describe the hypothesis, the domain, and its discretization. Then, the two-way parabolic wave equation model is introduced. In what follows, we assume a $\exp(j\omega t)$ time dependence and a slowly varying refractive index $n$. Moreover, with the studied frequency range, we assume no ground-wave propagation.

#### 2.1.1. Domain and Discretization

In this article, we study the propagation over the ground, which is at $z = 0$, along the x-direction. Thus, the usual 2D Cartesian coordinate system $(x, z)$ is used, with $z$ as the altitude. We assume that the source is placed at $x_s \leq 0$, and the field is thus computed in the domain $[0, x_{max}] \times [0, z_{max}]$. In this context, the field can be decomposed into transverse electric (TE) or transverse magnetic (TM) components. In this work, only the TE component

is studied, since the computations remain the same for the TM case. The TE part of the field is denoted by $\psi$ and is a solution of the Helmholtz equation.

Next, for obvious numerical reasons, the domain must be discretized. First, a sampling along the $z$-axis is performed as follows:

$$z[p_z] = \Delta z p_z \text{ with } p_z \in \{0, \cdots, N_z\}, \tag{1}$$

where $N_z$ corresponds to the number of discretization points, and $\Delta z = z_{\max}/N_z$ the vertical step. At a position $x$, the discrete version of a field $\psi$ at altitude $p_z$ is denoted by $\psi_x[p_z]$. Second, a mesh along the propagation direction $x$ is also performed with a step $\Delta x$ and a number of points $N_x$.

### 2.1.2. Parabolic Wave Equation Model

To study the tropospheric long-range propagation, a convenient model is the parabolic wave equation (PWE) [17]. By only accounting for the forward propagation in a paraxial cone along the propagation direction, this asymptotic model reduces the Helmholtz equation [16,17] as follows:

$$\frac{\partial u_f}{\partial x} = -jk_0\left(\sqrt{\frac{1}{k_0^2}\frac{\partial^2}{\partial z^2} + 1} - 1\right)u_f - jk_0(n-1)u_f, \tag{2}$$

with $u_f$ being the reduced field in the forward direction and $k_0$ the free-space wave number. Note that Equation (2) corresponds to the wide angle PWE [17], with a paraxial cone of almost $40°$. Nevertheless, one of the main limitations of this model is that it does not account for backward propagation. Thus, a two-way version of the PWE has been proposed in [24]. This latter is given by the system of equations:

$$\frac{\partial u_f}{\partial x} = -jk_0\left(\sqrt{\frac{1}{k_0^2}\frac{\partial^2}{\partial z^2} + 1} - 1\right)u_f - jk_0(n-1)u_f \tag{3}$$

$$\frac{\partial u_b}{\partial x} = jk_0\left(\sqrt{\frac{1}{k_0^2}\frac{\partial^2}{\partial z^2} + 1} - 1\right)u_b + jk_0(n-1)u_b, \tag{4}$$

where $u_b$ corresponds with the backward propagation term, as defined in [24–26], and only appears when reaching an obstacle, where reflections are introduced. It is important to note that the backward and forward equations are the same within the sign of $k_0$. Therefore, for these two equations, a two-way model is introduced, which allows us to take more precisely into account the reflection on obstacles along the propagation [26]. In the following sections, a numerical scheme to efficiently solve these equations is proposed.

### 2.2. Brief Reminder on the 1D Discrete Wavelet Transform

Since wavelets are at the center of the SSW propagation method, in this section, we provide a brief overview of the 1D multilevel discrete wavelet transform (DWT). For more information, the interested reader is referred to [33].

In the DWT, wavelets are used as the decomposition basis in place of the Fourier atoms for the Fourier transform. In a few words, wavelets correspond to short-length oscillating functions located both in space and frequency.

To perform the DWT, first a wavelet family is constructed. This family leans on a mother wavelet, denoted by $\psi$, of zero mean. This latter is then dilated on $L$ levels in order to cover the spectrum. Indeed, with $L$ increasing, the lower parts of the spectrum are covered. This function is also translated by $p$ to cover the spatial domain. The dilated and translated functions are part of the family denoted by:

$$\mathcal{F} = \left\{\psi_{l,p} \middle| (l,p) \in [1,L] \times \mathbb{Z}\right\}, \tag{5}$$

where $\mathbb{Z}$ corresponds to the set of relative integers. In order to cover the lowest part of the spectrum and the continuous part, the scaling function $\phi_{L,p}$ of non-zero mean is added to the family. Thus, an orthonormal basis is obtained. An example of the spatial and spectral coverage of this multi-resolution basis is pictured in Figure 1. In Figure 1a, the spatial coverage of wavelets of each level is shown, while in Figure 1b the spectral coverage for each level is plotted.

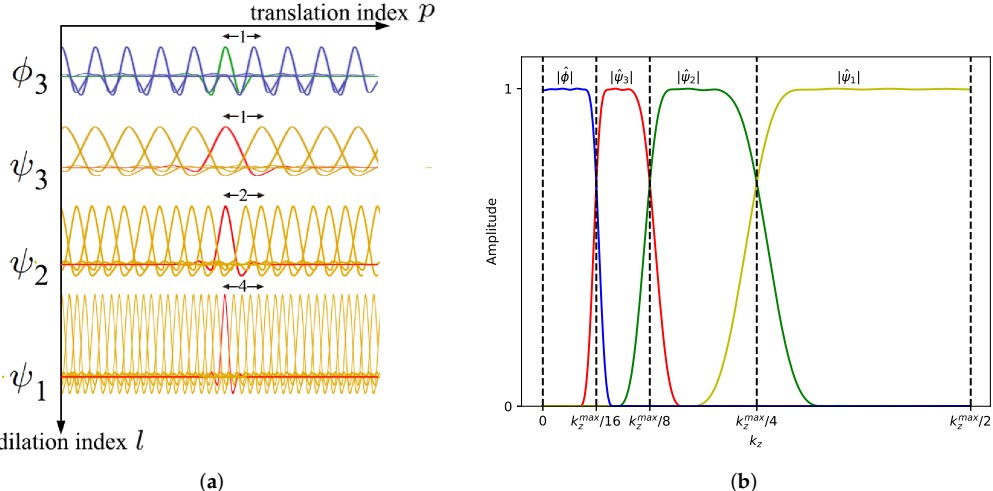

(**a**)                                                                 (**b**)

**Figure 1.** Example of a wavelet basis. (**a**) In the spatial domain. (**b**) Spectral coverage.

We can now decompose the reduced field on this basis through the DWT [33] as follows:

$$u_x[\cdot] = \sum_{p=0}^{N_z/2^L-1} a_{L,p}\phi_{L,p}[\cdot] + \sum_{l=1}^{L} \sum_{p=0}^{N_z/2^l-1} d_{l,p}\psi_{l,p}[\cdot]. \tag{6}$$

In this equation $a_{L,p}$ represents the approximation coefficients and corresponds to the decomposition on the scaling function. The details coefficients, denoted by $d_{l,p}$, correspond to the decomposition of the field on the family $\mathcal{F}$. Finally, $N_z/2^l - 1$ corresponds to the number of coefficients for each level; thus, $N_z$ must be a power a of 2 for the SSW method. In the rest of this paper, the wavelet decomposition is denoted by **W**.

To compute the approximation and details coefficient, an efficient method is the fast wavelet transform (FWT) [33]. This latter is of complexity $O(N_z)$, lower than the FFT.

An important property of the wavelets for the FWT is the number of vanishing moments, $n_v$. A smooth signal is described with fewer coefficients, with $n_v$ increasing [33]. Thus, few coefficients are needed to describe the field, and they mostly describe its discontinuity [33,35]. A compression, denoted by $\mathbf{C}_V$, with hard threshold $V$ is thus applied on the decomposition only to keep the important information. Finally, different wavelet families can be used for the decomposition. Here, the symlet family, which is almost symmetric with $n_v = 6$, and a maximum level of decomposition $L = 3$ are chosen. For more information about these choices, the reader is referred to [28].

### 2.3. Two-Way SSW over Rough Surfaces

In this section, the computational method is developed. First, a reminder of the one-way SSW algorithm is provided. Second, its generalization to solve the two-way PWE is introduced. Third, we remind the reader of the hybrid approach of [34] to treat rough sea surfaces and incorporate them to SSW. Finally, a theoretical comparison of the complexity and the memory usage between SSF and SSW is proposed.

### 2.3.1. Overview of the One-Way SSW Scheme

Before describing the two-way scheme, a brief reminder of the conventional one-way SSW method [29] is provided. This latter is an iterative method that computes the propagation, marching in on distances from the source. A step of propagation from $x$ to $x + \Delta x$ is described through the following four steps:

1.  The FWT, operator $\mathbf{W}$, and the compression operator with threshold $V_s$, denoted by $\mathbf{C}_{V_s}$, are applied to the reduced field $u_x$ to obtain a sparse set of wavelet coefficients:

$$U_x = \mathbf{C}_{V_s}\mathbf{W}u_x. \tag{7}$$

Thus, only the coefficient of the field higher than $V_s$ is kept, leading to a faster propagation.

2.  The wavelet coefficients are propagated through a free-space layer from $x$ to $x + \Delta x$ using the sparse wavelet-to-wavelet propagator, denoted by $\mathbf{P}$:

$$U_{x+\Delta x} = \mathbf{P}U_x. \tag{8}$$

Contrary to SSF, where a diagonal operator is used for the free-space propagation of plane wave, here, a wavelet-to-wavelet propagator is needed. Thus, we are required to use the method described in [29] to compute the propagation step in the wavelet domain. In a few words, a minimal number of wavelet propagations on one step are computed using the SSF scheme. The sparse wavelet decomposition of these local propagations, using an FWT and compression with threshold $V_p$, is then stored in a set of local propagators. This latter is pre-computed but can be computed again throughout the propagation if needed [29]. After that, this set is used to obtain all the local propagations associated with all of the non-zero wavelet coefficients of the field, which are then summed to obtain the vector of the propagated wavelet coefficients $U_{x+\Delta x}$.

3.  The free-space propagated field is then obtained through an inverse FWT as follows:

$$u_{x+\Delta x}^{fs} = \mathbf{W}^{-1}U_{x+\Delta x}. \tag{9}$$

4.  Finally, the effects of the environment are accounted for in the spatial domain. In particular, the refraction effects from the atmosphere are computed through a phase-screen operator [16], denoted by $\mathbf{R}$, as follows:

$$u_{x+\Delta x} = \mathbf{R}u_{x+\Delta x}^{fs}. \tag{10}$$

The operator $\mathbf{R}$ is a diagonal and accounts for the refraction at each step. Its elements are defined as:

$$\mathbf{R}[p_z, p_z] = \exp(-jk_0(n[p_z] - 1)\Delta x). \tag{11}$$

These steps have been described assuming no ground conditions. To account for the ground effects, the local image method [30] is used here. As a matter of fact, given the local aspect of the wavelets, the local image method allows us to consider the ground with only $N_{im} \ll N_z$ more coefficients, whereas adding $N_z$ coefficients would be needed with the usual image method. Indeed, a local replica of the field multiplied by the reflection coefficient $Z_0$ is generated. Then, the total field corresponding to the field in the computational domain and in the thin image layer is propagated in one step. Then, only the field in the computational domain is kept. Thus, by choosing $N_{im}$ wisely, no parasite reflections reach the computational domain. Note that this latter value is calculated as the maximum support of the wavelets after one step of propagation. Since the support of a wavelet is very small compared to the domain size, this method is thus efficient. Finally, the relief is considered through the staircase model [17].

### 2.3.2. Generalization to the Two-Way SSW

In this part, the generalization to solve the two-way PWE is introduced. Briefly, it corresponds to applying the one-way SSW algorithm by switching back and forth between the forward and backward propagations when reaching obstacles.

First, as mentioned before, the forward field $u_f$ is propagated with a step $\Delta x$ using the one-way SSW. Second, the backward field $u_b$ propagated along the axis $x$ with a step $-\Delta x$. Since the only difference between (3) and (4) is the sign of $k_0$, and, furthermore, the sign for the propagation step is also changed between forward and backward propagations, the propagator remains the same for both $u_f$ and $u_b$ [26].

Therefore, when reaching an obstacle, the backward field is initiated and propagated toward the opposite direction using the same one-way SSW computational scheme. This technique allows us to consider more accurately multiple reflections and the multi-path effect with the PWE [24], but the computation time and the memory load are increased.

As mentioned above, the retro-propagated field needs to be initiated. To do so, we use the conditions at the obstacle position $x_o$. Since the staircase model is used, the condition on the transverse component gives the following equation at the obstacle:

$$\exp(jk_0x_o)u_f(x_o) + \exp(-jk_0x_o)u_b(x_o) = t\exp(jk_0x_o)u_f(x_o), \tag{12}$$

with $t = \sqrt{1-r^2}$ as the transmission coefficient and $r$ as the Fresnel reflection coefficient. Note that if the transverse condition of the obstacle corresponds to a PEC, then $t = 0$ and the reflection is the total as mentioned in [24,26]. Thus, when reaching an obstacle, the previous equation allows us to compute the initialization of the retro-propagated field. Note that, since the staircase model is used here, the corner diffraction is ignored [26].

Then, to reduce the amount of computations and thus, the computation time, a stopping criterion must be introduced [25,26]. Using the compression introduced through the wavelet decomposition, we show that the stopping condition is intrinsic here, differing from [25,26] and using an advantage of the wavelet transform.

To prove this proposition, we use properties of the wavelet decomposition [33,36]. First, let us introduce the operator norm, which is defined as:

$$\|\mathbf{P}\|_{\mathrm{op}} = \sup_{u \neq 0} \frac{\|\mathbf{P}u\|_2}{\|u\|_2}. \tag{13}$$

Second, if the propagation is performed in free space, with not-evanescent waves and no boundaries, then we have:

$$\|\mathbf{P}u\|_2 = \|u\|_2, \tag{14}$$

since the energy remains the same in the domain. Otherwise, the energy can only leave the domain, in the apodization area for example, or remain constant, thus in all generality we have:

$$\|\mathbf{P}\|_{\mathrm{op}} \leq 1. \tag{15}$$

Note that this result is straightforward considering the $1/\sqrt{r}$ decrease of the field magnitude (2D Green's function). Therefore, it follows for the reduced field that:

$$\forall n \geq 0, \|u_n\|_2 \leq \|u_0\|_2. \tag{16}$$

This means that propagated fields always have a norm less or equal to the previous fields and, in particular, the initial field. Next, the Moyal relation [33] is used to obtain a condition on the wavelet coefficients of the field as follows:

$$\forall n \geq 0, \|U_n\|_2 \leq \|U_0\|_2. \tag{17}$$

Now recall that a compression with a hard threshold is performed at each step with SSW. The threshold is expressed as follows:

$$V_s = v_s \|U_0\|_\infty. \tag{18}$$

Thus, the normalized threshold $v_s$ has the same effect as the stopping criterion used in [26]. Moreover, throughout the propagation, this latter value can be changed to reduce the number of backward propagations and thus the computational time. Additionally, a theoretical formula has been obtained to assess the error of compression of SSW with the number of iterations [37].

In conclusion, in SSW the two-way generalization is performed by propagating the backward field initiated with the condition with the one-way SSW method (12) when reaching an obstacle. The stopping criterion here is implicit with the compression performed on the wavelet coefficients. This numerical method is used for large-scale obstacles, such as knife-edge or relief, to account for the multi-path effect and reflections; it is not used for small scale obstacles, such as sea waves, where this effect is negligible.

### 2.3.3. Introduction of Rough Surfaces

In this section, we introduce the hybrid method to model a rough sea surface of [34] in SSW.

The main idea of the method is to also consider the sea surface geometry and not just an attenuation through a roughness parameter in the reflection coefficient.

To consider the sea effect, one uses the sea spectrum, denoted by $S_z$, such as the Pierson–Moskowitz [38] one or the Elfouhaily [39] one. An example of this latter one is pictured in Figure 2. In this paper, the latter is considered, since it is the more accurate regarding the experimentation data. When considering snowy clutter, a Gaussian spectrum [40] is used as in [41]. In a normal approach, this spectrum is used to compute the roughness coefficient with the Ament [42] or Miller and Brown [43] ones. Thus, roughness surfaces are only accounted for through an attenuation coefficient, and no shadowing effects are considered. In the hybrid method [34], the sea spectrum is divided into two parts. The lowest part allows us to compute the geometry of the sea surface, while the higher part is used to calculate a new roughness coefficient. In the following, we only use the Miller and Brown one, since this is more accurate.

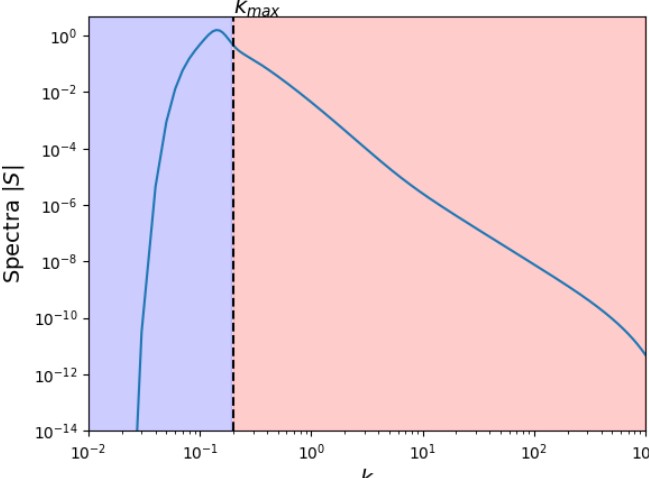

**Figure 2.** Splitting of the Elfouhaily spectrum. The red and blue parts correspond to the high and low roughness parts, respectively.

For better readability, the method is explained for rough sea surfaces, but the approach remains the same for any other rough surfaces, such as a snow clutter. Firstly the Elfouhaily spectrum is divided in two from the propagation parameters. The division limit is given by:

$$k_{\max} = N_x \frac{2\pi}{x_{\max}}. \tag{19}$$

This limit is shown in Figure 2.

First, we consider the part below the limit $k_{\max}$, in blue in Figure 2. This lower part of the spectrum is used to generate a random sea surface as follows. A random altitude profile is generated as a Gaussian white noise vector $B$ of size $N_x$. Then, this latter is convolved to the inverse Fourier transform of the square root of the sea spectrum, $\sqrt{S_z}$ corresponding to sea waves, to obtain a random sea surface geometry along the propagation axis $x$. Note that, for efficiency, the convolution is performed as a product in the spectral domain. Thus, the sea surface geometry $z$, corresponding to the altitude at each point on the axis $x$, is given by:

$$z(x) = \mathbf{F}^{-1}\left(\sqrt{S_z}\mathbf{F}(B(x))\right), \tag{20}$$

with $\mathbf{F}$ being the Fourier transform operator and $\mathbf{F}^{-1}$ its inverse. Therefore, using (20) random sea surface geometries are generated. Second, the higher part of the spectrum is used to compute the attenuation coefficient as follows, see the red part of Figure 2. The new standard deviation of the low roughness waves is computed with:

$$h_{sc} = 4\sqrt{\int_{k_{\max}}^{\infty} S_z(k)dk}. \tag{21}$$

This is used to compute a new Miller and Brown roughness coefficient $\rho$ [34], which is defined as:

$$\rho = \exp\left(\frac{\gamma_r^2}{2}\right) I_0\left(\frac{\gamma_r^2}{2}\right), \tag{22}$$

with $\gamma_r = 2kh_{sc}\sin(\alpha)$, where $\alpha$ is the grazing angle, and $I_0$ is the modified Bessel function of order 0. The coefficient $\rho$ corresponds to an attenuation due to the roughness of the sea surface. Finally, the reflection coefficient $Z$ for the local image method is computed as:

$$Z = \rho Z_0, \tag{23}$$

with $Z_0$ being the Fresnel reflection coefficient, see Section 2.3.

This method allows us take both the geometry of the surface, the shadowing effects due to the waves, and the roughness and attenuation of the sea into account. This latter factor can also be used to consider terrain roughness with the Gaussian spectrum [40,41]. Note that the surfaces are randomly generated, thus Monte Carlo simulations are used to obtain the effect of the sea on the field in given conditions.

### 2.3.4. Comparison of SSW and SSF

In this section, a complete comparison between SSF and SSW in terms of complexity and memory usage is performed.

First, we denote by $N_s$ and $N_p$ the number of non-zero coefficients of $U_x$ and $\mathbf{P}$. They correspond to:

$$N_s = \sharp\{U_x[i]/\ \forall 0 \leq i \leq N_z,\ U_x[i] \neq 0\}, \tag{24}$$

$$N_p = \sharp\{\mathbf{P}[i,j]/\ \forall(i,j) \in [0,N_z]^2,\ \mathbf{P}[i,j] \neq 0\}. \tag{25}$$

Given that the signals we are dealing with are smooth functions, these numbers are very low compared to $N_z$ and $N_z^2$, respectively. Moreover, they can be approximated through the formula given in [35]. Using this, and since the signals we are dealing with are smooth, one can see that, for example, $N_s$ is of the order of 10 coefficients. Therefore, in practice with an appropriate threshold we have:

$$N_s \ll N_z \text{ and } N_p \ll N_z. \tag{26}$$

This result has been validated through numerical simulations in [28].

Second, we can compare the memory usage of both methods for each propagation step. In SSF, we need to store the diagonal operator of propagation; thus, we have $N_p^{SSF} = N_z$. We also need to store the spectral transformation of the field corresponding to $N_s^{SSF} = N_z$. Thus, in terms of memory efficiency, SSW is better than SSF, given that a good compression is performed.

Third, the complexities of both methods are compared for one step of propagation. On one hand, the complexity of the SSF method corresponds to the sum of the complexity of the FFT, the propagation step, and the inverse FFT. This leads to a complexity of:

$$\mathcal{O}(N_z \log(N_z)) + \mathcal{O}(N_z) + \mathcal{O}(N_z \log(N_z)) = \mathcal{O}(N_z \log(N_z)). \tag{27}$$

On the other hand, the complexity of SSW corresponds to:

$$\mathcal{O}(N_z) + \mathcal{O}(N_s N_p) + \mathcal{O}(N_z) = \mathcal{O}(N_s N_p). \tag{28}$$

Thus, since $N_s N_p \leq N_z$, when a good compression is applied, the complexity of SSW is also lower than the one of SSF.

Therefore, SSW is theoretically better than SSF in terms of both memory and time efficiency, which is useful in our context. Indeed, Monte Carlo simulations are needed to compute the effects of the sea on the propagation. It is also necessary for the generalization to 3D [31,32].

## 3. Results

In this section, numerical simulations are performed. First, we validate the one-way and two-way SSW computational schemes. Second, a propagation test with knife-edge obstacles in the S-band is performed to compare the results to [25,26]. Next, the SSW scheme with the hybrid method is validated with two different scenarios: propagation in a maritime environment and propagation over snowy clutter. Finally, the method is applied on different problematic scenarios, such as the prediction of a radar coverage, optimization of an antenna location, and as the direct method in the RFC context. All these tests are performed at different frequency ranges.

### 3.1. Validation of SSW

In this section, we aim to validate one-way SSW by comparing the results to the exact solution for a complex source point (CSP) [13].

Therefore, the propagation from a CSP at $f_0 = 300$ MHz is studied along the $x$ axis. The computations are performed in a domain of size $x \in [0, 4000]$ m and $z \in [0, 2048]$ m. The steps are $\Delta x = 50$ m and $\Delta z = \lambda/2 = 0.5$ m along the $x$ and $z$ axes. The source is placed at $x_s = -50$ m and $z_s = 1024$ m with a width of $W_0 = 5$ m. For comparison with the exact solution, we assume $n = 1$, and the domain has been defined such that the propagation can never reach the ground. The thresholds in SSW are set so as to obtain a maximum compression error of $-20$ dB at the end.

The results are plotted in Figure 3a–c. In the first image, Figure 3a, we have plotted the propagation obtain with SSW. Figure 3b shows the difference between SSW and the exact solution on the computational domain. The last, Figure 3c, pictures the fields at the last iterations obtained with SSW and the exact solution.

In Figure 3b, it can be seen that the error between the exact solution and SSW is below $-20$ dB, which is negligible, as expected. Moreover, the error is mainly outside a cone along the propagation direction, which is due to the paraxial approximation, the compression introduced in SSW, and also to the mesh size as mentioned in [44]. Moreover, one can see in Figure 3c that both fields are matching on the last iteration until $-50$ dB. Thus, the one-way SSW scheme works well.

Furthermore, in this case, the propagation for 80 iterations has been performed in 1 s, and the memory size of the propagator is of the order 1 kB, showing that the method is also efficient. Further studies have been carried out in [28,44] to validate the method; these show the effects of the different parameters.

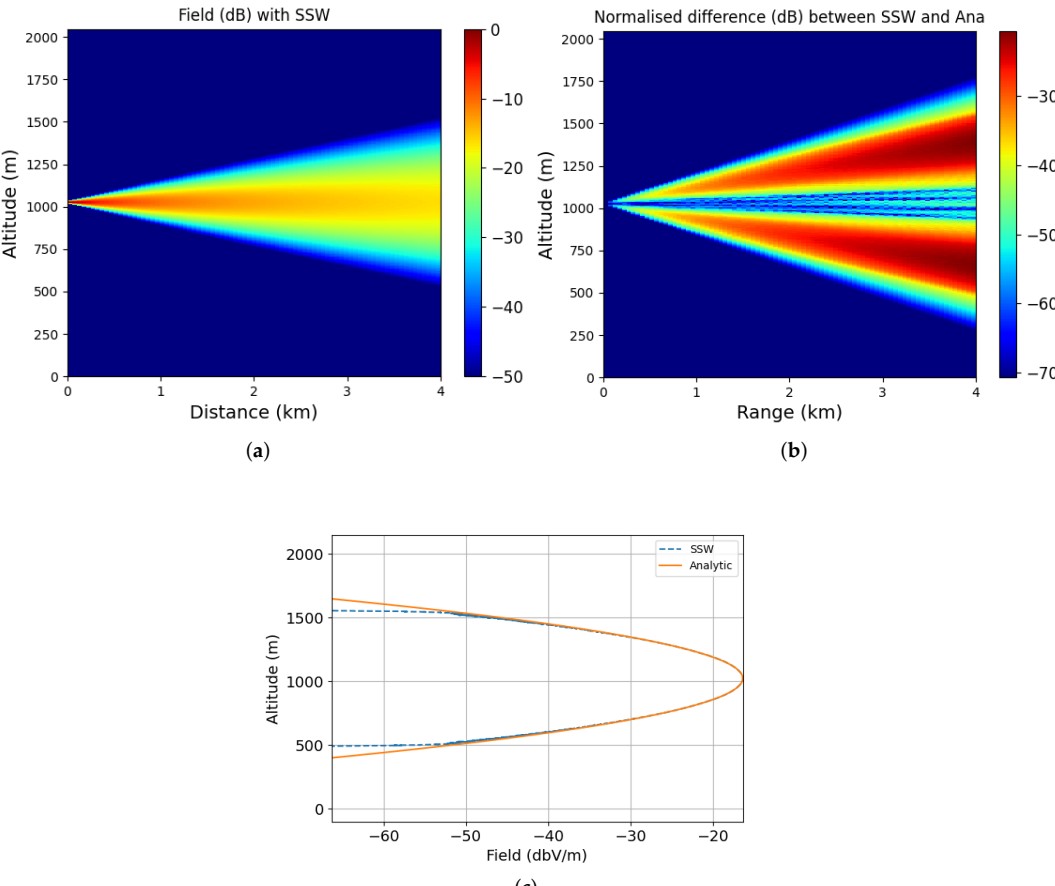

**Figure 3.** Propagation in free space for comparison with the exact analytical solution of the CSP. (**a**) Normalized field obtained with SSW. (**b**) Normalized difference between SSW and the analytical solution. (**c**) Comparison of the exact solution and of the field obtained with SSW at the last iteration. (**a**) Normalized reduced field $u$ (dB) obtained with SSW. (**b**) Normalized difference between SSW and the exact solution. (**c**) Exact solution and field obtained with SSW at the last iteration.

In conclusion, the one-way SSW works well and is efficient in terms of both computation time and memory usage.

### 3.2. Validation of the Two-Way SSW

In this section, we validate the two-way SSW method. To do so, a plane wave at $f_0 = 300$ MHz (UHF-band) is propagated in the x direction until a PEC of the size of the computational domain is reached. We expect that the total field will be negligible.

For this scenario, the domain is of size $x_{\mathrm{max}} = 1000$ m and $z \in [0, 1024]$ m. The mesh size is $\Delta x = 50$ m along the $x$-axis and $\Delta z = \lambda/2$ along the $z$-axis. A PEC wall of 1024 m is placed at $x_o = x_{\mathrm{max}}$. Until this obstacle, we propagate in free-space. Thus, an apodization window below and above the computational domain is used to avoid parasite reflections. In addition, to validate the two-way method, we assume $n = 1$ to only account for the propagation scheme. We plot the normalized total field in dB, sum of the incident and reflected fields, in Figure 4.

As can be seen in Figure 4, the total field is negligible in the propagation domain, as expected. Indeed, at the PEC wall, where the reflections begin, it is below $-60$ dB, while increasing to below $-35$ dB at the origin of the domain. Thus, the two-way version of SSW works well in a canonical test.

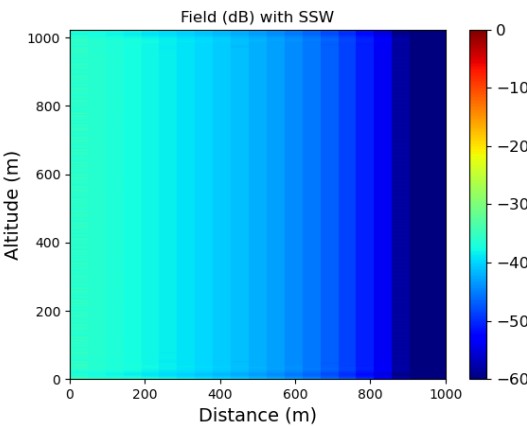

**Figure 4.** Normalized reduced field (dB) obtained with SSW.

### 3.3. Propagation over Two Knife-Edge Obstacles

Now that two-way SSW has been validated, the method is used to compute the propagation over the ground, while considering two knife-edge obstacles, as in [25,26].

Here, the propagation from a complex source point [13] in the S-band, $f_0 = 3$ GHz, is studied. The source is placed at $x_s = -50$ m and $z_s = 50$ m with a width of $W_0 = 5$ m. The computation domain is $(x, z) \in [0, 60{,}000] \times [0, 512]$ m$^2$. The steps along both axes are $\Delta x = 200$ m and $\Delta z = 0.1$ m. To validate the method in this case, a PEC ground condition is considered, and we assume $n = 1$. We also consider two knife-edge reliefs placed at 20 km and 40 km at altitude 100 m and 150 m, respectively. Here, the thresholds are set using the theoretical formula in [37], such that a maximum error of $-20$ dB with SSF is obtained. Note that the thresholds also correspond to the implicit stopping criterion of SSW.

The normalized reduced field obtained with the two-way SSW method is plotted in Figure 5b. The results for the one-way version of SSW are also picture in Figure 5a in order to compare both results.

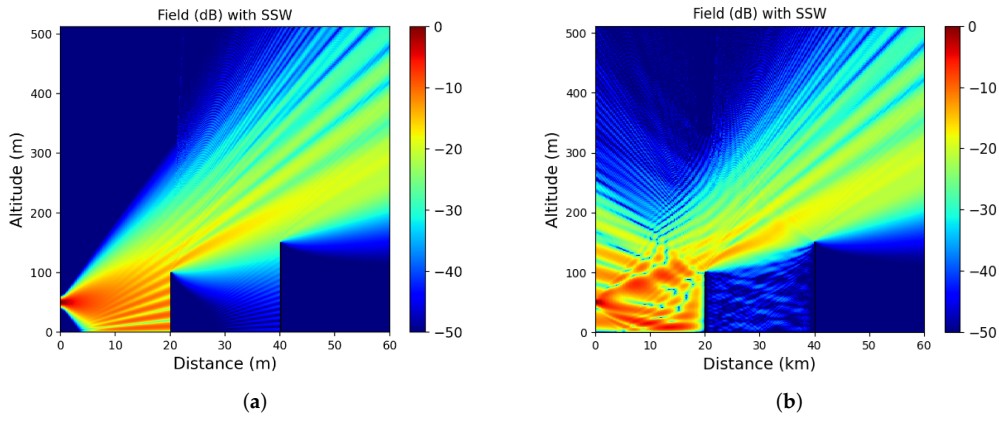

(**a**) (**b**)

**Figure 5.** Propagation of the normalized field $u$ (in dB) computed with the one-way and two-way SSW, respectively. (**a**) One-way SSW. (**b**) Two-way SSW.

First, the error between the one-way SSW and SSF is below $-45$ dB, as expected. Second, with the one-way method, see Figure 5a, only the forward propagation is computed. Thus, the reliefs induce only shadow areas and diffraction in the propagation direction. As can be seen in Figure 5b, with the two-way algorithm, we also consider the reflections due to the relief. Therefore, the multi-path effect in between the relief is considered here, but the computation time is increased to account for all the backward propagations. Note that only the implicit stopping criterion has been used here, even if the multiple reflections between both obstacles are considered. Additionally, the results are in line with those

obtained in [26], showing that the method works well. Therefore, if we want to accurately compute the coverage of a given antenna while accounting for complicated structures, the two-way version is better. Nevertheless, if the computation is limited, for example in the RFC inversion problem, or if only the last iteration is needed, we can use the one-way version of SSW.

*3.4. Propagation above the Sea*

In this section, we validate the one-way SSW method with the hybrid approach for the propagation in a maritime environment. The results are thus compared to the ones obtain with SSF [34,41].

In this scenario, we model the propagation from the Saint-Mathieu Lighthouse (Plougour-den in France) and the airport of Ouessant (France) at $f_0 = 9$ GHz (X-band). The considered source is a CSP placed at $(xs, zs) = (-50, 2)$ m, and its width is $W_0 = 2$ m. The relief between the source and ending points is obtained through the data provided by the "Institut Nationale de l'Informations Géographique et Forestière" (IGN) [45]. Thus, the islands between both places are also considered, such as the island of Molène.

The computational domain is of size [0, 26,000] × [0, 123] m². The mesh sizes are $\Delta x = 50$ m and $\Delta z = 0.03$ m along the $x$ and $z$ axes, respectively. For the different ground conditions, we consider the parameters of a dry ground ($\varepsilon_r = 20$ and $\sigma_r = 0.02$ S/m), for the terrain, and of the water ($\varepsilon_r = 80$ and $\sigma_r = 5$ S/m), for the sea surface. We generate the sea surface geometry using the hybrid approach described in Section 4. A wind speed of $U_{10} = 10$ m/s is considered. We also consider an evaporation duct at the sea surface [46]. The wavelet parameters remain the same for this test.

The results are plotted in Figure 6. Figure 6a shows the propagation of the reduced field u computed with SSW. In Figure 6b, we show the normalized difference between SSW and SSF along the propagation.

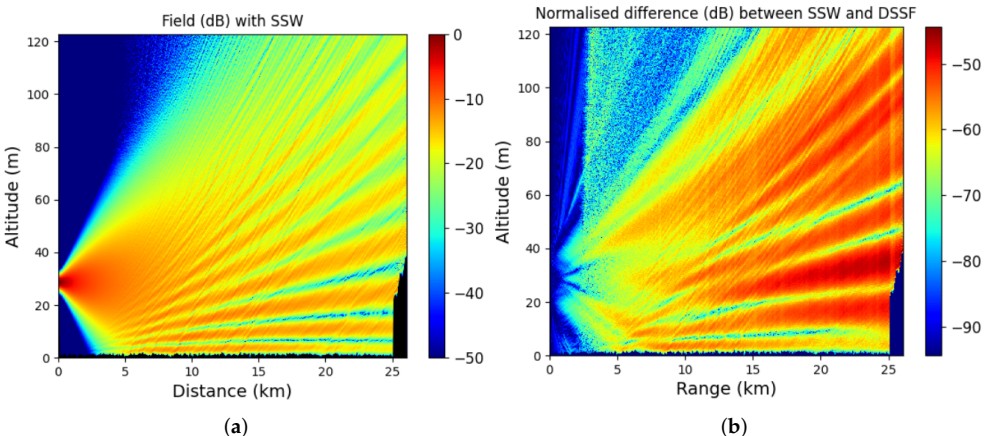

**Figure 6.** Propagation above a rough sea surface. (**a**) Propagation of the normalized field *u* (in dB). (**b**) Normalized difference between SSW and SSF.

First, in Figure 6, one can see that we account for both the effects of the refraction and of the sea surface geometry. Indeed, the electromagnetic waves are straight-lined near the sea, in the surface duct. In addition, one can also note the effects of the surface geometry with the diffractive pattern in the propagation. Second, the error is below −45 dB. Thus, the SSW with the hybrid approach is validated in this case. Furthermore, the results are in line with those obtained in [34,41]. Finally, only one simulation has been performed here. Since the sea surface generation is random, a Monte-Carlo approach should be considered and is performed in Sections 3.7 and 3.8.

To conclude, we note that using this approach allows us to model more precisely the effects of the sea. This is of high importance for the RFC inverse problem [2,3], where an accurate and fast forward model is needed. This latter is studied in Section 3.8 for various

sea conditions. This is also of serious concern for the prediction of radar coverage and optimization of antenna location near the sea, as will be seen in Section 3.7.

### 3.5. Two-Way Propagation in Snowy Condition

In this section, we test the SSW method while considering snowy ground conditions [41].

In this scenario, we study the propagation from a CSP in the UHF-band ($f_0 = 500\,\text{MHz}$) over a snowy ground. The snow dielectric parameters are taken from [47], so as to compute the reflection coefficient for the local image method. Therefore, we have $\varepsilon_r = 30$ and $\sigma_r = 3 \times 10^{-5}\,\text{S/m}$. The hybrid method is used to take into account both the snow surface and the attenuation. A triangular relief is also considered.

The source of the parameters remain the same, except that the source altitude is $z_s = 70$ m. Finally, the computations are performed in the following domain: $(x, y) \in [0, 15,000] \times [0, 308]\,\text{m}^2$. The mesh sizes along $x$ and $z$ are $\Delta x = 50$ m and $\Delta z = 0.3$ m. The propagation is computed both with the one-way and two-way SSW methods. For the two-way method, to avoid unnecessary computations, backward computations are only computed for reliefs of more than 2 m in altitude. Finally, the wavelet parameters remain the same.

The results are plotted in Figures 7 and 8, for the propagation computed with one-way and two-way SSW, respectively. The normalized reduced field on the overall domain is shown.

First, one can note the interest of using the hybrid approach. Indeed, the effect of the roughness of the snow clutter can be seen as an interference pattern due to the relief introduced by the snow surface. Second, for both methods, as expected, the results beside the triangular relief are the same. Nevertheless, in front of the relief, the two-way method allows us to consider the reflection toward the source. Thus, we have obtained an accurate propagation method in various environments.

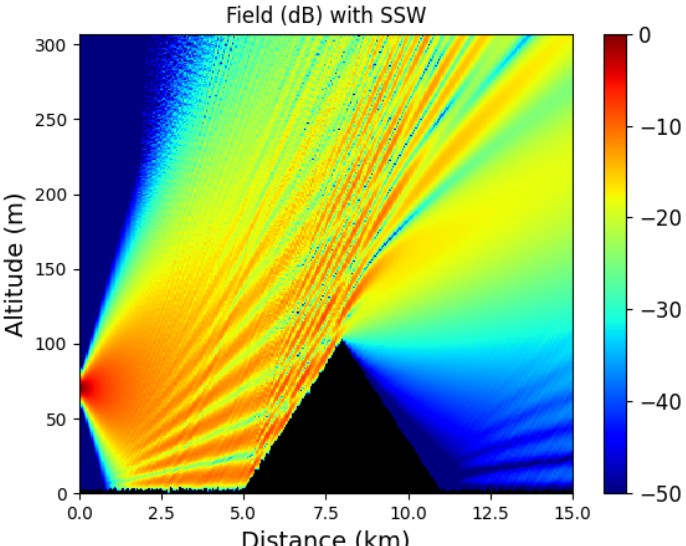

**Figure 7.** Propagation over a rough terrain obtained with one-way SSW.

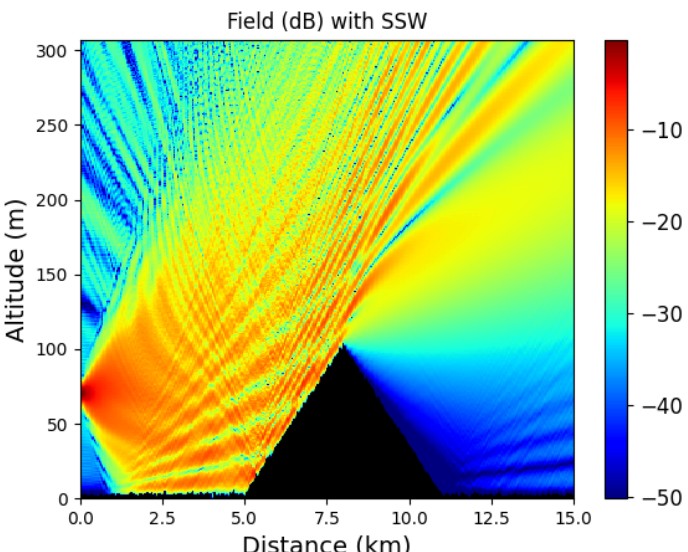

**Figure 8.** Propagation over a rough terrain obtained with two-way SSW.

### 3.6. Application to the Prediction of Radar Coverage

In this numerical experiment, the two-way SSW method is applied to predict the coverage of the Toulouse airport radar (France) in one direction.

To do so, we model the antenna propagation pattern as a CSP [13] in the VHF band (at $f_0 = 300$ MHz). The parameters of the source are as follows: $x_s = -50$ m, $z_s = 10$ m and $W_0 = 5$ m. The 2D propagation between the Toulouse airport and Montauban is modeled here with the two-way SSW method and the SSF scheme in order to compare them. The relief between the two places is taken into account using IGN data [45]. To account for a realistic effect, a tropospheric duct modeled with a tri-linear profile of refraction [46] is accounted for. The parameters for $M$ are $M_0 = 330$ M-units, $z_b = 300$ m, and $z_t = 250$ m, and the gradients are $c_0 = 0.118$ M-units/m and $c_2 = -0.5$ M-units/m; see Figure 9 for a presentation of the parameters.

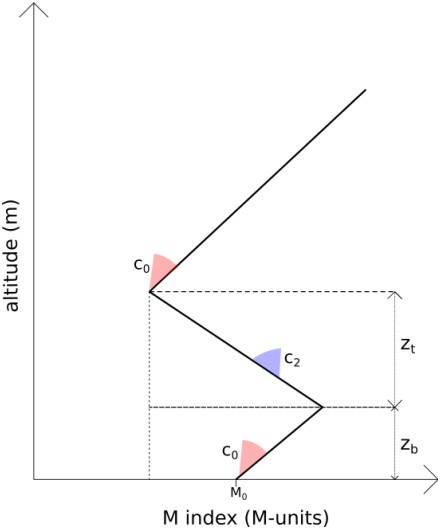

**Figure 9.** Tri-linear profile of atmosphere.

This latter models a tropospheric duct as a three layers refractive index that varies linearly in each layer with a different gradient. Thus, $z_b$ corresponds to the altitude until the first change of gradient from $c_0$, positive, to $c_2$, negative, and $z_t$ to the transition altitude, with $z_b + z_t$ the altitude of the second change of gradient from $c_2$ to $c_0$.

The computational domain is of size $x \in [0, 42{,}000]$ m and $z \in [0, 512]$ m and sampled with steps $\Delta x = 100$ m and $\Delta z = 0.5$ m. We consider a dielectric ground condition of parameters $\varepsilon_r = 20$ and $\sigma_r = 0.02$ S/m, which correspond to the conditions of a dry ground. Finally, the wavelet parameters remain the same.

For this scenario, we first plot in Figure 10a,b the field computed with the one-way and two-way schemes, respectively. Second, the normalized difference between both schemes is pictured in Figure 11.

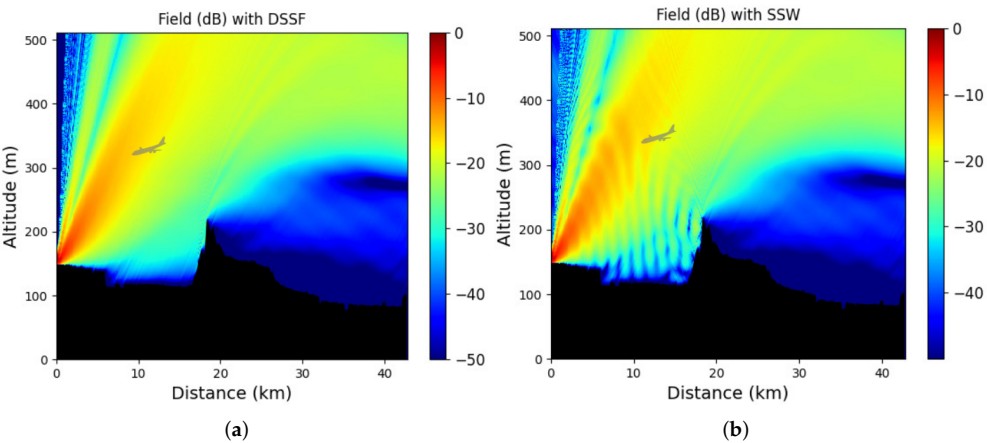

(a)  (b)

**Figure 10.** Prediction of the radar coverage for the Toulouse airport using the one-way and two-way methods. (**a**) One-way method. (**b**) Two-way method.

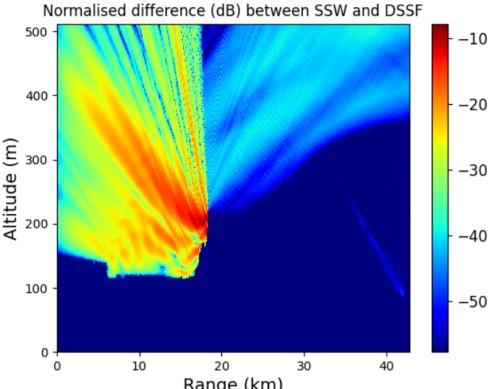

**Figure 11.** Normalized difference between the one-way SSF and two-way SSW schemes along the propagation.

Figure 10 shows the differences in the prediction of the radar coverage between the one-way and two-way methods. Indeed, in between both reliefs the two-way SSW scheme takes into account the backward propagation. Thus, an interference pattern appears, but no shadowy areas above 300 m appear due to the multiple reflections, which is very important in the context of radar coverage. Therefore, a landing plane, such as the gray one, is spotted by the radar. Otherwise, after 20 km, one can see that both methods give the same result. Moreover, the difference between both methods is small, as can bee seen in Figure 11. Thus, even with the reflections, an airplane can be spotted.

Note that, as pictured in Figure 11, the difference between both schemes is localized in between the relief, as expected, and is of order $-10$ dB, which is small. Moreover, the error after the second relief is due to the compression introduce in SSW. Therefore, the choice between the two-way and one-way schemes to predict radar coverage mostly depends on the environment. If more reliefs, such as metallic structures, are considered, then reflections are important and should be considered even if the computation time increases.

Nevertheless, this scenario shows that the method works well and is useful to compute radar coverage in given conditions.

### 3.7. Application to the Optimization of an Antenna Location

We now study the propagation in a maritime environment, as in [34,41]. In particular, we apply the SSW method with the hybrid approach for the sea to optimize the antenna location for given conditions.

For the following tests, the propagation is studied in the X-band with $f_0 = 9$ GHz. The source is placed in the harbor of Toulon (France). Thus, the propagation is modeled above the Mediterranean sea until 20 km from the source. The propagation domain is, thus, of size $x \in [0, 20,000]$ m and $z \in [0, 125]$ m. The considered source is a CSP placed at $x_s = -50$ m with a width of 2 m. The altitude of the source above the ground will vary along the numerical tests and is the parameter to be optimized here. The goal is to obtain the best possible coverage. The steps are as follows: $\Delta x = 50$ m and $\Delta z = 0.03$ m.

The sea dielectric parameters are $\varepsilon_r = 80$ and $\sigma_r = 5$ S/m [47]. Sea surfaces and the attenuation parameters are computed through the hybrid approach using the Elfouhaily spectrum. We consider a wind speed of $U_{10} = 10$ m/s. Since the surface generation is random, 50 Monte-Carlo simulations are performed.

We also consider a tropospheric duct above the sea, modeled with a tri-linear atmospheric index [46]. The parameters are as follows: $M_0 = 330$ M-units, $z_b = 20$ m, and $z_t = 50$ m, and the gradients are $c_0 = 0.118$ M-units/m and $c_2 = -0.5$ M-units/m. The wavelet parameters remain the same.

We first consider an antenna located $z_s = 5$ m above the ground. In this case, the predicted coverage is plotted in Figures 12 and 13. In these figures, we plot the mean of the sum of the 50 reduced fields obtained through the Monte-Carlo simulations and the worst-case scenario in terms of ship detection, respectively.

From the results pictured in Figures 12 and 13, we can conclude there is a shadowy area at 15 km where ships would not be spotted, as pictured with a gray ship that can not be seen. Thus, the coverage of this antenna needs to be improved. In order to do so, we change the antenna location to $z_s = 20$ m above the ground.

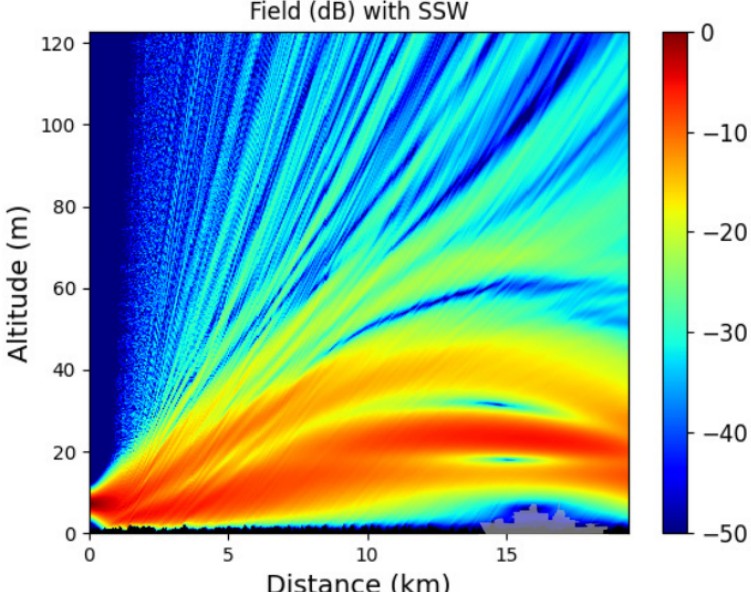

**Figure 12.** Mean reduced field over 50 Monte-Carlo simulations computed with the SSW method. In this case the source is placed $z_s = 5$ m above the ground.

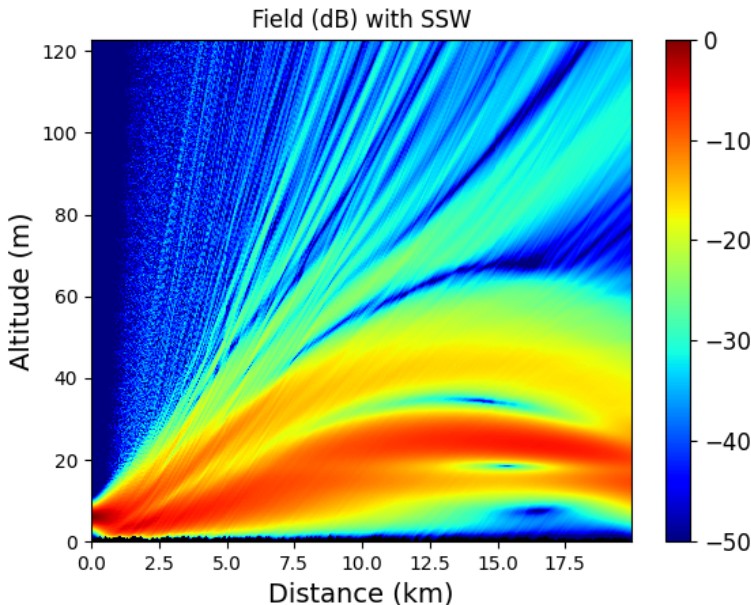

**Figure 13.** Worst-case scenario for the antenna coverage in terms of ship detection. In this case the source is placed $z_s = 5$ m above the ground.

As before, we plot in Figure 14 the mean of the sum of the field obtained through 50 Monte-Carlo simulations with the SSW scheme. As before, we also plot the worst-case scenario in Figure 15 to verify that a ship will still be detected.

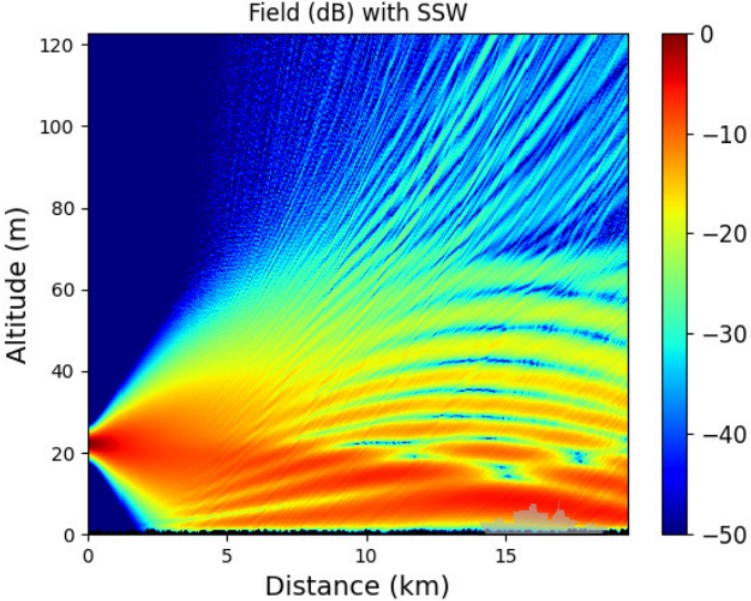

**Figure 14.** Mean reduced field over 50 Monte-Carlo simulations computed with the SSW method. In this case the source is placed $z_s = 20$ m above the ground.

The first conclusion from Figure 14 is that, by changing the antenna location, shadowy areas are no longer seen around the sea, even in the worst case scenario, see Figure 15. Therefore, any ship, such as the gray one, can be spotted in this case. Nevertheless, in this case, the propagation is more stuck in the tropospheric duct, and less energy exists above 60 m. This is not a problem in our context (ship detection).

This scenario shows that the method is also useful for the optimization of an antenna's location. Obviously more tests in different conditions should be performed to conclude

on the final antenna position. Furthermore, if the antenna altitude can be changed, this method allows us to find a good position given the location conditions.

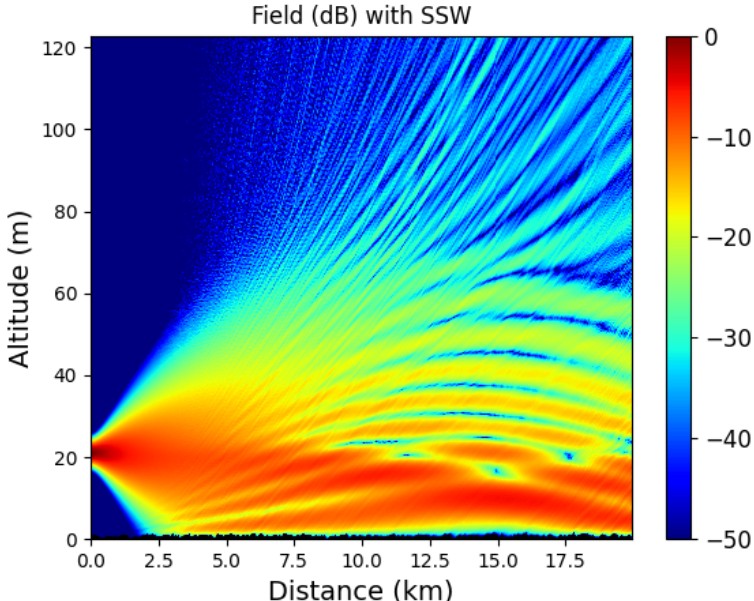

**Figure 15.** Worst-case scenario for the antenna coverage computed with the SSW method. In this case the source is placed $z_s = 20$ m above the ground.

*3.8. Forward Model for the RFC Problem*

In this last numerical test, we study the propagation above the sea from the antenna of a ship, which corresponds to the forward model of RFC [2,3]. The main objective is to show that the surface geometry must be taken into account in this context. Thus, different conditions of wind to generate the sea surface are studied. Moreover, in the RFC scenario, the goal is to have a fast forward model, thus we use the one-way SSW scheme.

Here, the propagation above the sea from a CSP at $f_0 = 9$ GHz (in line with the frequency used in RFC [48]) is modeled. The sea surface is generated through the hybrid approach of Section 2.3 for different wind speeds $U_{10} \in [5, 10, 15, 20]$ m/s.

For all wind conditions, the following parameters are the same. The source is located at $x_s = -50$ m and $z_s = 10$ m above the ground (on the top of the ship). Its width is $W_0 = 2$ m. The computational domain is of size $(x, y) \in [0, 20,000] \times [0, 123]$ m $\times$ m. The discretization is performed with steps $\Delta x = 50$ m and $\Delta z = 0.03$ m. We consider an impedance ground condition of parameters $\varepsilon_r = 80$ and $\sigma_r = 5$ S/m that corresponds to the parameters of water. We also consider a surface duct of 20 m. The wavelet parameters are the same as in the previous tests. Finally, for each wind speed, 50 Monte-Carlo simulations are performed.

In Figure 16a–d, we plot the means of the propagated field over all Monte-Carlo simulations for the different wind speeds $U_{10} \in [5, 10, 15, 20]$ m/s, respectively.

Figure 16 shows the effects of different sea surfaces geometry on the propagation of electromagnetic waves. Indeed, for different wind speeds the sea waves are stronger, thus the propagation is affected. As can be seen, the radiation lobes throughout the propagation are different between the four figures (a) through (b). As a matter of fact, the more the wind speed increases, the fewer lobes are presents, but the first lobe becomes larger. As can be seen in Figure 16d, we only have one lobe for a high wind speed. In the RFC scenario [2,3], we measure the field at the last iteration in order to retrieve the refractive index. Thus, if the sea geometry is not accounted for, one can see that error would be introduced in the forward model.

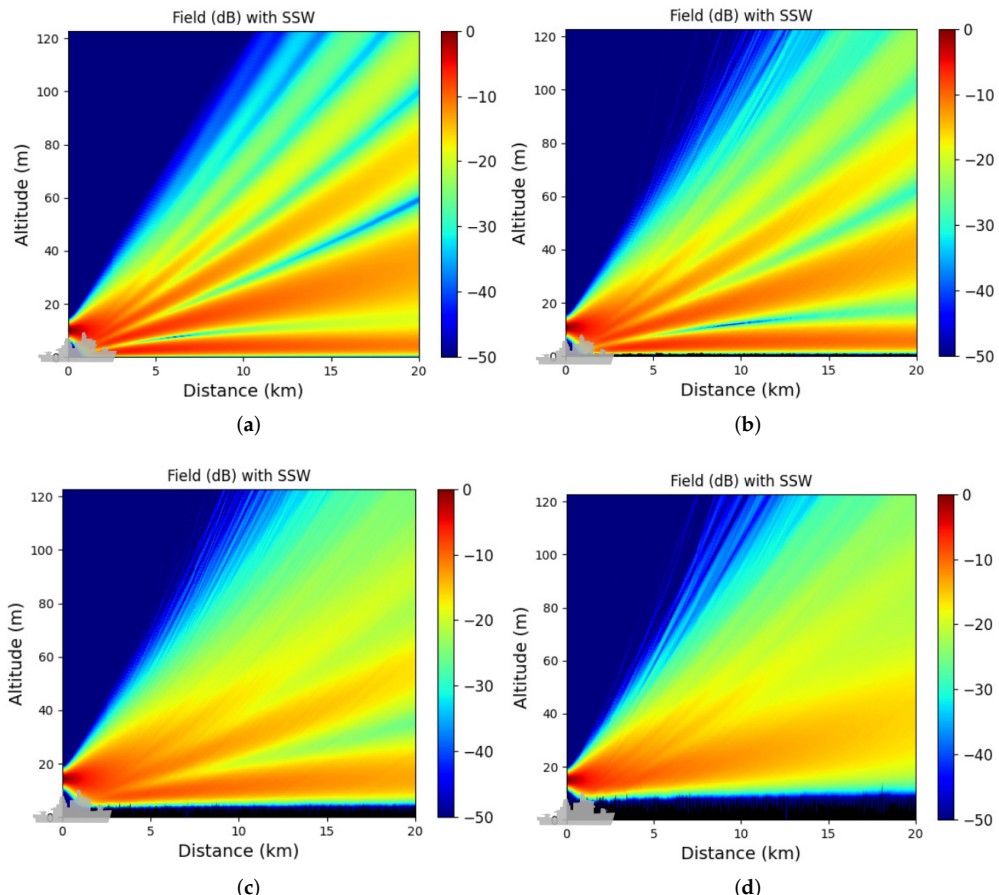

**Figure 16.** Means of the propagated fields over 50 Monte-Carlo simulations for different wind speeds. (**a**) $U_{10} = 5$ m/s. (**b**) $U_{10} = 10$ m/s. (**c**) 15 m/s. (**d**) 20 m/s.

## 4. Discussion

Characterizing the propagation canal is very important for many applications, such as radar, teledetection, or communications. In this context, one needs to model the long-range propagation in the troposphere while accounting for the relief and the ground composition and roughness. Thus, fast, accurate, and memory-efficient computational methods are needed. To improve the accuracy of the existing SSW method, we have introduced here a two-way version of SSW in order to consider back-propagations. Furthermore, a hybrid approach to consider rough surfaces has been introduced to SSW. In addition, this method is better both in terms of memory efficiency and computation time than the usual SSF method.

First, the two-way PWE has been introduced. This allows us to take into account both the forward and backward propagations, while the usual PWE only considers the forward part. Second, the SSW scheme is introduced to solve the two-way PWE. We show that no explicit stopping criterion is needed, since the compression introduced in the wavelet decomposition works as an implicit stopping criterion. A complete comparison between SSF and SSW in terms of time and memory efficiency is proposed. This shows that, with good compression, SSW is better than SSF in both parameters. Finally, numerical experiments are performed. They allow us to validate the method and show that two-way SSW works well in various conditions (relief, sea, snow) and at various frequencies (UHF-band, S-band, X-band). Applications of the method in different scenarios, such as the optimization of an antenna location or the prediction of radar coverage, are also proposed.

We have thus shown that the newly developed two-way SSW is efficient for modeling the tropospheric long-range propagation in various environments and useful for different applications. Indeed, the low complexity of the wavelet transform and the compression

introduced in SSW allow us to obtain an accurate and a memory- and time-efficient numerical scheme. Additionally, using the wavelet properties, the stopping criterion for backward propagation is implicit and can be changed throughout the propagation, departing from the two-way SSF. This adds versatility to the method. Furthermore, ground conditions and roughness are considered with this scheme, with no cost on the computation time.

Nevertheless, this method has limitations. First, the complexity and memory usage rely mostly on the compression. Thus, if the signal we are dealing with is not smooth, then the computation time increase. In addition, taking into account the backward propagations with the two-way SSW scheme increases the computation time. Thus, given the scenario and the accuracy needed, a choice between the one-way or two-way computational method must be performed. Secondly, the method solves the wide-angle PWE, and the results are valid in a cone of around $40°$. Third, the reliefs considered are limited in slope, since the method is (for now) only developed for the staircase model. More reliable models of terrain are under study, but this induces a change in the wavelet-to-wavelet propagator and in the initial condition for the two-way SSW method. Finally, some physical phenomena are not considered, such as the diffusion or complex interaction of the electromagnetic waves with the sea waves.

Further works include more numerical tests. This could also lead to a basis of numerous computed propagations for different inputs (ground condition, sea surface geometry, refraction, etc.), which could be useful for inverse-problem or artificial intelligence-based propagation schemes [49]. We are also investigating how to efficiently parallelize the two-way version of SSW. Furthermore, other applications of this method should be studied, such as the ionospheric propagation with SSW [50]. We are also investigating a hybridization of SSW with a wavelet-based method of moments (MoM) [51], similar to SSF with the MoM [52], for a better accuracy when dealing with relief and man-made structures. To conclude, the generalization of the approach to 3D [31,32] would be an evolution of the proposed work.

**Author Contributions:** Conceptualization, T.B. and O.B.; methodology, T.B. and O.B.; validation, T.B.; investigation, T.B.; writing—original draft preparation, T.B.; writing—review and editing, O.B. and A.K. All authors have read and agreed to the published version of the manuscript.

**Funding:** This research received no external funding.

**Data Availability Statement:** Data were not used or created for this research.

**Conflicts of Interest:** The authors declare no conflict of interest.

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
