# Peer review of "A Two-Way Split-Step Wavelet Scheme for Tropospheric Long-Range Propagation in Various Environments"

_remotesensing, doi:10.3390/rs14112686_

Round 1
Reviewer 1 Report
The manuscript has presented a two-way split-step wavelet (SSW)scheme for tropospheric long-range propagation in various environment. Some comments are as following:
1. It is suggested to give clearer description for equations (7),(12),(18).
2. It is suggested give clearer description for the figure 8.
3. It is suggested to pay more attention to the writing. For example, these sentences are hard to understand: “Since the support of a 150 wavelet is very small compared to the domain size is method is efficient.” (line150-151) and “To so a plane wave at f0 = 300 MHz (UHF-band) is propagated in the x-direction until a PEC of the size of the computational domain.”(line231-233).
4. It is suggested to provide actual example to validate presented method in practical way besides the numerical tests.
Author Response
Dear reviewer,
First of all we thank you for your comments and relevant questions, so you will find our answers via the downloaded file.
All the best

Reviewer 2 Report
1. The parabolic equation method developed by the authors is approximate. Unfortunately, only various versions of the parabolic equation method are compared in the paper and there is no comparison with the exact solution or with other alternative asymptotic methods.
2. It is known that diffraction at sharp edges produces an edge wave (V.A. Borovikov, B.Ye Kinber Geometrical Theory of Diffraction Institution of Electrical Engineers, 1994). How is its influence taken into account?
3. Calculation algorithms are not described in sufficient detail. Many important issues are presented in the form of links to other already published sources, which makes it difficult to understand the text of the work.
4. From the work it is difficult to understand whether the algorithms developed in the work can be applied to real-time calculations.
5. When applying the Monte Carlo method, average values ​​are used. Maybe it's better to focus on the worst-case scenario?
6. There are stylistic repetitions in the work. You can edit the text a little.
7. The authors write (241-243) that “As can be seen in Figure 3, the total field is negligible in the propagation domain, as 241 expected. Indeed, at the beginning it is below -60 dB while increasing to below -35 dB at 242 the end". From fig. 3 it is logical to assume that the origin is on the left (where 0) and not on the right.
Author Response

(The authors gave the same response as above.)
